# Neck Lymph Node Recurrence in HNC Patients Might Be Predicted before Radiotherapy Using Radiomics Extracted from CT Images and XGBoost Algorithm

**DOI:** 10.3390/jpm12091377

**Published:** 2022-08-25

**Authors:** Yi-Lun Tsai, Shang-Wen Chen, Chia-Hung Kao, Da-Chuan Cheng

**Affiliations:** 1The Ph.D Program for Medical Engineering and Rehabilitation Science, College of Biomedical Engineering, China Medical University, Taichung 404333, Taiwan; 2Department of Radiation Oncology, China Medical University Hospital, Taichung 404327, Taiwan; 3School of Medicine, College of Medicine, China Medical University, Taichung 404333, Taiwan; 4School of Medicine, College of Medicine, Taipei Medical University, Taipei 110301, Taiwan; 5Center of Augmented Intelligence in Healthcare, China Medical University Hospital, Taichung 404327, Taiwan; 6Department of Nuclear Medicine and PET Center, China Medical University Hospital, Taichung 404327, Taiwan; 7Graduate Institute of Biomedical Sciences, School of Medicine, College of Medicine, Elite Campus, China Medical University, Taichung 404333, Taiwan; 8Department of Bioinformatics and Medical Engineering, Asia University, Taichung 41354, Taiwan; 9Department of Biomedical Imaging and Radiological Science, China Medical University, Taichung 404333, Taiwan

**Keywords:** radiomics, machine learning, XGBoost, head and neck cancer, recurrence

## Abstract

The five-year overall survival rate of patients without neck lymph node recurrence is over 50% higher than those with lymph node metastasis. This study aims to investigate the prognostic impact of computed tomogram (CT)-based radiomics on the outcome of metastatic neck lymph nodes in patients with head and neck cancer (HNC) receiving definitive radiotherapy or chemoradiotherapy for organ preservation. The pretreatment 18F-FDG PET/CT of 79 HNC patients was retrospectively analyzed with radiomics extractors. The imbalanced data was processed using two techniques: over-sampling and under-sampling, after which the prediction model was established with a machine learning model using the XGBoost algorithm. The imbalanced dataset strategies slightly decreased the specificity but greatly improved the sensitivity. To have a higher chance of predicting neck cancer recurrence, however, clinical data combined with CT-based radiomics provides the best prediction effect. The original dataset performed was as follows: accuracy = 0.76 ± 0.07, sensitivity = 0.44 ± 0.22, specificity = 0.88 ± 0.06. After we used the over-sampling technique, the accuracy, sensitivity, and specificity values were 0.80 ± 0.05, 0.67 ± 0.11, and 0.84 ± 0.05, respectively. Furthermore, after using the under-sampling technique, the accuracy, sensitivity, and specificity values were 0.71 ± 0.09, 0.73 ± 0.13, and 0.70 ± 0.13, respectively. The outcome of metastatic neck lymph nodes in patients with HNC receiving radiotherapy for organ preservation can be predicted based on the results of machine learning. This way, patients can be treated alternatively. A further external validation study is required to verify our findings.

## 1. Introduction

Every year worldwide, more than 830,000 patients are diagnosed with head and neck cancer (HNC), and more than 430,000 patients die because of the disease [1], which ranks head and neck cancer sixth or seventh among the world’s most common cancers. The overall HNC mortality rate is significant, ranging from 40% to 50% [2]. According to Surveillance Epidemiology and End Results (SEER) data, the five-year survival rate for all stages of HNC is approximately 60%.

Although there are many new treatment options to improve the overall survival rate (OS), the number of patients with local recurrence (LR) and distant metastasis (DM) is still high [3]. Risk stratification is especially important in order to provide patients with more precise treatment [4]. In addition to considering the primary tumor of HNC, lymph nodes should also be considered as an important outcome to determine if there is a risk of recurrence. In a study from 2018, it was found that patients with lymph node metastasis, which are neck recurrence (NR) in HNC, have a five-year survival rate of 37%, while those without them have a five-year survival rate of 74%. The OS of patients without NR is 50% higher than those with lymph node metastasis. Therefore, neck failure is an important indicator of patient survival after treatment [5]. Selective local lymph node dissection, extensive lymph node removal, or prophylactic neck radiation therapy may be performed in order to reduce the risk of recurrence and spread to ipsilateral or bilateral lymph node sites. Individualized treatment plans should be based on tumor location and lymph node status [6].

Therefore, it is particularly important to quickly and effectively develop an appropriate treatment plan for the patient. In addition to clinical information, image information with artificial intelligence analysis can also be used to improve the probability of choosing the best treatment [7].

Physicians use 18F-FDG PET/CT scans for TNM staging to provide patients with treatment plans. However, we observed that there were more biomarkers to be gathered from images; therefore, we used radiomics to extract data from 18F-FDG PET/CT images. Radiomics as a method is growing rapidly in research and can extract hundreds of quantitative features from medical images. A study in 2016 titled ’Radiomics: images are more than pictures, they are data’ [8], pointed out that radiomics was designed to distinguish details within medical images that were insensible to human eyes. These features might have the potential to provide clinically useful information [9].

As most radiomic analyses focused on the analysis of primary tumors [10,11], there are no studies using lymph nodes to predict HNC patients having neck recurrence (NR), which is an important indicator of survival. Moreover, neck recurrence is a key indicator for the development of personalized therapy. As mentioned before, the five-year survival rate of patients without cervical recurrence is more than double of those with NR. The purpose of this research was to develop a model based on the combination of image features extracted by radiomics and clinical data, which has the potential to predict the probability of neck recurrence in HNC patients in advance of treatment planning and thus might lead to more appropriate treatment.

## 2. Materials and Methods

Given the purpose of this study, we utilized several methods which included the Lifex program to mask 18F-FDG PET/CT scans, radiomics to extract biomarkers from CT scans, over-sampling and under-sampling techniques to reduce data imbalance, and the XGBoost algorithm to predict neck recurrence. There is a more detailed introduction as follows and the workflow can be seen in Figure 1.

### 2.1. Patient Population

There were 110 patients who were part of an organ preservation program in the China Medical University Hospital from March 2007 to November 2013. They had received pre-treatment 18F-FDG PET/CT (Figure 2A,B) for RT planning or preconditioning staging. The eligibility criterion with respect to neck lymph nodes was seen in 18F-FDG PET/CT scans, and the images were readable in the LIFEx software version 7.0.0 (Institut Curie Centre De Recherche, Centre Universitaire, Bâtiment 101B, Rue Henri Becquerel, 91898 ORSAY CEDEX, 91400, Orsay, France). There are two reasons why we downsized the patient population. First, the CT/PET images of 17 patients could not be loaded into the LIFEx segmentation software. This was most likely due to an incompatible format. We attempted to obtain the data with the compatible format; however, the data we received was the only data available. Second, 14 patients only had primary tumors, with no obvious lymph nodes that could be seen in the images. Therefore, this study included 79 HNC patients. The patients consisted of 78 males and 1 female. The age range was between 37 to 78 years old, with a median age of 51. There were 36 patients with oropharyngeal cancer (OPC) and 43 patients with hypopharyngeal cancer (HPC). Only patients with OPC and HPC are included in this study. This is because the base of the tongue is generally categorized as oropharyngeal cancer, whereas mobile tongue cancer belongs to cancers of the oral cavity, and patients are always treated with surgery with or without adjuvant radiotherapy. In patients with buccal or lip cancer, radiotherapy is mainly used for adjuvant treatment. SUVmax (maximum standard uptake value in PET images had a minimum value of 2.2 and a maximum value of 30.6. The necrotic lymph nodes (LN)s in this study were defined in pretreatment contrast-enhanced CT scans and confirmed by radiologists. Due to the lack of a consistent consensus about extra nodal spread, this parameter was not analyzed in this study. Thirty patients experienced LR, 22 patients experienced NR, and 13 patients experienced distant metastasis (DM). In our cohort, the median follow-up duration for live patients was 69 months (36 months for the whole population). The relapse patterns are shown in Figure A1 and Figure A2. Most neck or local residual or recurrent tumors were found within 2 years after the completion of radiotherapy [12]. The overall survival can be seen in Figure A3. This study was approved by The Institutional Review Board (certificate number of local IRB, DMR99-IRB-010 (CR-11) and CMUH106-REC3-119 (CR-3)). Information about the 79 patients is shown in Table 1.

### 2.2. PET/CT Images

The patients were scanned using a PET/CT scanner (PET/CT-16 slice, Discovery STE; GE Medical System, Milwaukee, WI, USA). They were instructed to fast for at least 4 h before the administration of 18F-FDG, and FDG PET/CT imaging was conducted approximately one hour after the administration of 370 MBq of 18F-FDG. The patient data we used for this research are from March 2007 to November 2013. At the time, the radiologists administered 370 MBq = 10 mCi of 18F-FDG universally. CMUH radiologists adjusted the dose based on the weight only after 2015. Both PET and CT scans with lymph node masks were analyzed using a machine learning model, and we then compared their *p*-value. Furthermore, we used the 18F-FDG PET/CT images and clinical data of the research subjects as research data, and the PET images were used to delineate the cancer area.

### 2.3. Tumor Region Delineation

LIFEx is a free Java application used to manually delineate the tumor boundaries using 2D or 3D drawing tools and the volume of interest (VOI) of tumors could be used as input for radiomics extractors. LIFEx-7.0.0 was used in this study. First, we loaded the CT scans as the background layer after which we loaded the PET scans to reveal the standardized uptake value (SUV), and then areas of the primary tumor and lymph nodes were highlighted with two colors using the 3D drawing tool, respectively (Figure 2C). We set the lower limit of SUVmax at 2.0 and then delineated the range of its uptake area. We set the lower limit at 2.0 for two reasons: (1) the clinical data of this study showed that the minimum SUVmax was 2.2; (2) a study suggested that in patients with primary tumor SUVmax > 2.5 and nodal SUVmax 2.0 to 6.0, SUV was more accurate in predicting lymph node malignancy [13]. Primary tumors were marked in pink, while lymph nodes were marked in yellow based on Lengele’s criteria [14]. All segmentation of the cancerous regions (VOI) was carefully confirmed by an experienced physician (S.-W. Chen, 30 years of experience). Afterward, CT scans with masks were extracted with a radiomics extractor.

### 2.4. Radiomics Extractor Selection

Radiomics is a method that can extract features from image textures to find hidden clues [15]. The term radiomics was first coined by Lambin et al. (2012) to describe quantitative medical imaging data. Radiomics involves extracting high-throughput data from medical images such as CT, PET, MRI, or SPECT scans through advanced mathematical and statistical analysis. Radiomic has several types of features: shape-based features, first-order statistics, Gray Level Run Length Matrix (GLRLM), Gray Level Co-Occurrence Matrix (GLCM), Gray Level Size Zone Matrix (GLSZM), Neighboring Gray Tone Difference Matrix (NGTDM), Gray Level Dependence Matrix (GLCM), A Laplacian of Gaussian (LoG) features, and Wavelet features.

Radiomics explores the intensity, shape, and texture of the tumor in order to calculate thousands of advanced features. It contains eight feature extractors: example_allShape, exampleCT, exampleMR_3mm, exampleMR_5mm, exampleMR_NoResampling, exampleVoxel, Params and MR_2D_extraction. In this study, the main data consisted of CT scans; therefore, we used the exampleCT extractor to extract the data from the images.

### 2.5. Imbalanced Dataset

Most medical image datasets were imbalanced since the probabilities of diseases were disproportionate. This imbalance caused a problem in training data. It also happened in our dataset. The total number of patients in this study was 79. There is a significant gender bias present. The incidence of human papillomavirus (HPV)-associated oropharyngeal cancer was very low in many Asian countries like Taiwan (<20% in our study cohort). More than 90% of our studied patients had a history of smoking, alcoholism, or betel nut squid, which is regarded as a carcinogen. On the other hand, the incidence of HPV-associated hypopharyngeal cancer was also very low in Asian countries (<10% in our study cohort). According to cultural habits, very few women had a history of smoking, alcoholism, or betel nut chewing. This was the reason that the gender for most of the studied subjects was male. Furthermore, there is another bias in the form of an uneven ratio of patients with recurrence and those without. The number of patients with recurrence and non-recurrence were 22 and 57, respectively. We used two strategies to deal with imbalanced data and tried to find a better prediction performance. In the first strategy, we split the patient data into training and test datasets, with a ratio of 0.7 and 0.3, respectively. Then, a random over-sampling technique [16] was applied to replicate the recurrence (the minority) in the training set in order to balance the data, and then conduct training. An example is shown in Figure 3A.

The second strategy was to randomly choose the data (for both training and test sets) using recurrence and non-recurrence with a fixed ratio (1:1); the remaining patient data was discarded. This strategy was similar to the under-sampling technique [17]. An example is shown in Figure 3B.

### 2.6. XGBoost Algorithm

The XGBoost is the abbreviation for eXtreme Gradient Boosting, which was published in 2016 [18] with the purpose of classifying data. This model is optimized by building hundreds or even thousands of trees. In addition, each tree is co-related in order to reduce the training loss of the previous tree. XGBoost is very different from Random Forest which generates the final result by independent voting; however, XGBoost influences the final result jointly. Moreover, especially when it comes to imbalanced datasets, XGBoost showed better performance than Random Forest [19]. After utilizing the over-sampling technique and under-sampling techniques, we used the XGBoost algorithm to analyze the data which was separated into a training set (70%) and a test set (30%). The analysis was run one hundred times, each time with randomly chosen sets of patients.

Furthermore, the XGBoost algorithm has hyperparameters that can be adjusted to increase the elasticity of the model. This study adjusted and compared different combinations of learning_rate, n_estimators, max_depth, and subsamples.

## 3. Results

### 3.1. Selection of Radiomics Extractors

After using the eight extractors from radiomics to extract features from the 18F-FDG PET/CT images which were used to predict neck recurrence, we found that by analyzing PET images with lymph node masks there were 105 parameters in total with *p*-value < 0.005. By analyzing the CT images with lymph node masks, we found 258 parameters in total with *p*-value < 0.005. We found that radiomics derived from CT was superior to that of PET images for predicting neck lymph node recurrence. Therefore, the subsequent analysis used CT scans with image features extracted by the example CT extractor combined with the clinical data for machine learning.

### 3.2. Machine Learning Algorithms: Random Forest and XGBoost with Hyperparameter Selection

We originally ran the analysis using the Random Forest algorithm; however, the performance values were unsatisfactory. Therefore, we chose to use XGBoost which offered better results. The difference in the results by using the two different algorithms can be seen in Table 2.

In the machine learning hyperparameters section of XGBoost, in order to achieve the best results possible, we tuned the hyperparameters several times. We tuned the learning rate (0.01, 0.05, 0.1, 0.2), the number of trees (n_estimators—300, 400, 500), and the depth of the trees (max_depth—10, 20) and got the following results (Figure 4A,B). It can be seen from the chart that the model is the most stable when the hyperparameter is set to 400_20 (n_estimators_ max_depth). Furthermore, we included a third hyperparameter: the analysis of the learning rate, which suggested that 0.1 and 0.2 were more stable.

Regarding the issue of over-fitting, we adjusted the ‘subsample’ parameter to see its impact. The subsample is a built-in adjustable parameter in XGBoost, which is supposed to adjust overfitting [18,20]. We tested the subsample in three different ratios: 0.5, 0.8, and 1. The results are shown in Table 2. The following results were performed by using subsample = 0.8, which was a commonly used value [21]. This setting (subsample = 0.8) resulted in XGBoost randomly sampling 80% of the training data prior to growing trees, and this prevented overfitting to some extent and improved the model performance.

### 3.3. Over-Sampling and Under-Sampling

In order to mitigate data imbalance, we used random over-sampling to duplicate the training set of recurrence data one time (original ratio), two times, three times, and four times, then combined them with non-recurrence data, respectively, and then proceeded with machine learning. We found that when the data was replicated three times, the sensitivity was improved by reducing the specificity, and the performance was better (Figure 5).

For example, with a hyperparameter value of 400_20, the sensitivity value was 0.44 ± 0.22 in the original ratio of data. The value of specificity was 0.88 ± 0.06;w however, after duplicating the minority data three times, the sensitivity was raised to 0.67 ± 0.11 and specificity to 0.84 ± 0.05 (Table 3).

The random under-sampling method was used to treat both the training and test data sets with the same amount of recurrence and non-recurrence patients. As we can see, the highest sensitivity for predicting neck recurrence in data with the original ratio was 0.53, and the lowest was 0.36 (Table 3); however, after using the under-sampling strategy, the highest sensitivity value raised to 0.76, and the lowest was 0.69 (Table 4). Moreover, the results show that when we changed different parameters of the machine learning model, i.e., the number of trees (300, 400, 500) and depth (10,20), the sensitivity of the data was better than the data without under-sampling (Figure 6), regardless of the changes in the hyperparameters (Table 4).

### 3.4. Predicting Ability of Different Combinations of Data

An additional issue we were concerned about was whether there is a difference between clinical data alone or clinical data combined with lymph nodes radiomics data, or clinical data with lymph node and primary tumor radiomics data. The clinical data consists of 72 categories including age, gender, smoker/non-smoker, TNM staging system, nodal-metabolic tumor volume, total lesion glycolysis, gross tumor volume, and more. The radiomics data of both primary tumor and lymph nodes consist of 1256 features each, and the results are shown in the following figure (Figure 7). We can see that the clinical data with lymph node information is the best predictor of neck recurrence, followed by clinical information alone, and clinical information with lymph node and primary tumor radiomics data is the least effective.

### 3.5. The Important Features of Predicting Neck Recurrence

Although the average values of accuracy, sensitivity, and specificity were around 0.7, there were always 3–4 extreme outliers when running the XGBoost model 10 times or more. Therefore, we chose those three models which offered the best results, then analyzed the selected features and used Shap [22] which can be used to interpret the results from the XGBoost model to understand the relationship between features, and the results which affected neck recurrence the most.

As can be seen in Figure 8, the accuracy, sensitivity, and specificity of Figure 8A,B were 0.86, 1.00, and 0.71, respectively. After calculating the feature importance of this model, feature 531 ‘original_firstorder_Median’ had the greatest impact on the prediction model. For another model (Figure 8C,D), all values of accuracy, sensitivity, and specificity were 0.86, and the most impactful feature of this model was feature 8 ‘Lesion site (1) Opx (2) HPx’. The value of accuracy, sensitivity, and specificity in the last model (Figure 8E,F) was 1. This model also showed the best performance. The feature 1037 ‘wavelet-LHH_glszm_GrayLevelNonUniformityNormalized’, which also appeared in the previous models, was one of the key radiomics parameters for predicting neck recurrence. The feature names can be found in Table A1.

## 4. Discussion

This study aimed to use PET/CT-derived radiomics for predicting radiotherapy-based outcomes in patients with head and neck cancer. It has a very different purpose from other studies, that is, instead of using the primary tumor as the resource of our main data and lymph node to complement said data to predict local relapse, in this study, we used the lymph node as the main data set to predict neck cancer recurrence.

Since local control of head and neck cancer is generally good in the context of medical advancements, this is not proportional to the improvement in survival, precisely because of the development of neck recurrence—a key cause of failed treatment and subsequent mortality. Therefore, in order to improve the survival rate and prognosis of patients, it is important to determine the relevant prognostic factors to better assess the aggressiveness of the tumor at the time of diagnosis and personalize the treatment modalities. If physicians can predict that there will be a recurrence in the neck, they can increase dosage at the time of treatment, reducing the chance of recurrence and providing a better prognosis.

In addition, there is hidden information in medical images, which have potential relationships that the human eyes cannot see. Therefore, extracting data from images through a radiomics extractor not only increases variables but also provides more information that cannot be found manually. However, before being processed in the radiomics extractor, the volume of interest (VOI) must be calibrated in the image-reading program, which is a very time-consuming process. Therefore, in the future, it is expected that auto-segmentation can be used to significantly reduce the time needed for processing image data.

The global average of patients who are diagnosed with head and neck cancer is 9 people in 100,000. However, the number of patients diagnosed with head and neck cancer in Taiwan is 32.5 people in 100,000, making it the highest ratio in the world (Taiwan Ministry of Health and Welfare). Furthermore, in Taiwan, head and neck cancer is the third most common cancer in men. The median age of death of patients with head and neck cancer is 59 years, which is 18.3 years less than the average life expectancy for men. One of the main causes of HNC in Taiwanese patients is betel nut, which is mostly consumed by males. As a result, the data used in this research is somewhat biased either in terms of recurrence rate or gender, and gender especially is uneven. Nevertheless, we believe that gender bias will not have an effect on the prediction performance, and this study will be of great help to physicians in making more personalized treatment plans.

Furthermore, the collection and processing of data that consists of medical images usually take a long time, especially when recurrence is the purpose of the research, and the probability of disease recurrence cannot be fully understood. Therefore, medical research often encounters the problem of data imbalance. The analysis of this data has proven to be difficult, mainly due to the imbalance of patients with recurrence and those without, which is mentioned in the previous chapter; however, after utilizing the over-sampling technique, the values of accuracy and sensitivity have increased over the values from the original data set. In addition, when using the under-sampling technique, the value of sensitivity increased even further, compared to when using oversampling. The number of patients without recurrence was more than double of those with recurrence; therefore, over-sampling and under-sampling techniques were used to reduce the bias caused by data imbalance. It was found that both over-sampling and under-sampling slightly reduced specificity but substantially increased sensitivity [23]. We look forward to expanding the dataset and processing an open dataset for future research. Prospective implementation of data analysis across multiple centers may contribute to a better understanding of head and neck cancer and could help develop a model that can assist physicians in diagnosis.

Moreover, we believe that in any research, the more data we have available, the better, but sometimes that is not always the case. In addition, the data obtained from the images extracted by radiomics of the primary tumors was modeled with the same hyperparameters and the same proportion of cuts to predict neck failure. Clinical data combined with radiomics data has shown to have better predictive ability than clinical data alone, as expected. Surprisingly, combining primary tumor image information with lymph node information showed the worst predictive ability, with the same trend in terms of accuracy, sensitivity, and specificity.

Contrary to the results of studies that predict local recurrence in head and neck cancer, the combination of clinical information and radiomics information of primary tumors and lymph nodes is more effective than using primary tumor information alone [24]. We believe there are two possible reasons for this. First, local recurrence is defined as cancer that recurs in the same place as the original cancer or very close to it. Regional recurrence (neck recurrence in head and neck) means that the tumor has grown into lymph nodes or tissues near the original site of cancer. After extracting images with radiomics, both primary tumors and lymph have very different textures. Our results suggest that, in order to successfully predict neck recurrence, it is important to use lymph nodes as primary data. Consequently, we used lymph nodes to predict neck recurrence. Moreover, when we added the primary tumor data (used for predicting local recurrence), the performance in predicting neck recurrence decreased. The second possible reason is that the model used in this study utilizes the XGBoost algorithm. Too many parameters may reduce the probability of being selected for key parameters and fail to provide better predictions. This study had some limitations. First, external validation with independent data sets is essential to create the model’s clinical utility, because this study was conducted at a single institute. Second, although organ preservation with definitive radiotherapy is a mainstay of treatment for patients with cancers of hypopharynx or oropharynx, our findings should be interpreted cautiously because of the absence of other subsites of head and neck cancers. Nonetheless, this study represents a step toward enabling the customization of precision therapy for head and neck cancer patients when the optimal management of the metastatic neck node becomes an issue of debate. After further validation and inclusion of more tumor sites, oncologists may use the proposed model to advise patients on the relative suitability of treatment options.

## 5. Conclusions

This is the first study, to our knowledge, that utilizes image features extracted from lymph nodes to predict neck recurrence. This study suggests that not only clinical data but also data extracted from medical images provide many hidden clues that can be used to assist doctors in predicting recurrence. We also analyzed the importance of features after processing the data with machine learning, and we found several key radiomics parameters for predicting neck recurrence. The outcome of neck lymph nodes in patients with HNC receiving radiotherapy for organ preservation program can be predicted according to the results of machine learning by combining clinical data and CT-based radiomics of marked lymph node masks from pre-treatment 18F-FDG PET/CT images. The main purpose of this study was to see how information gathered from lymph nodes, rather the primary tumors, could be used to predict regional recurrence (neck recurrence in HNC). In this way, patients can be treated alternatively. Further external validation studies are required to verify our findings.

## Figures and Tables

**Figure 1 jpm-12-01377-f001:**
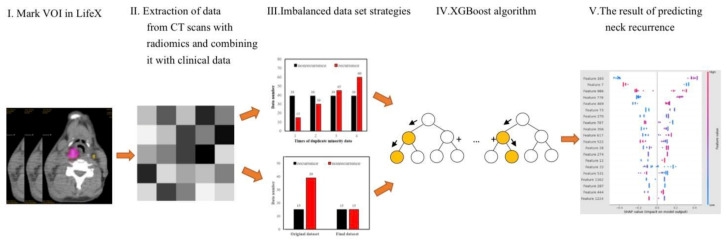
The study workflow.

**Figure 2 jpm-12-01377-f002:**
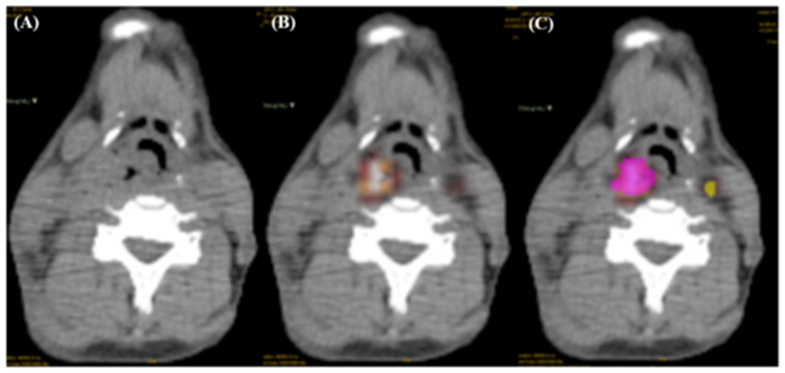
(**A**) CT scan (**B**) PET/CT scan (**C**) PET/CT scan with masks of primary tumor (pink) and lymph nodes (yellow).

**Figure 3 jpm-12-01377-f003:**
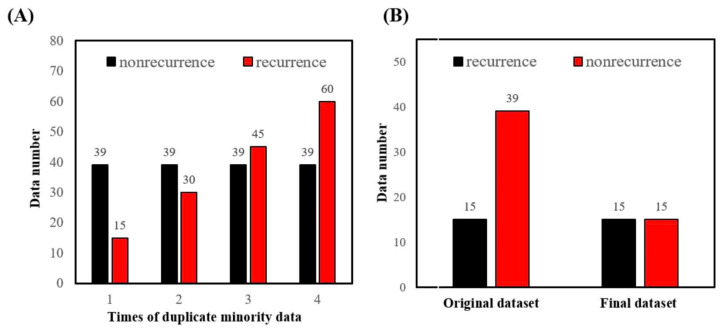
(**A**) Strategy 1: Random over-sampling replicated the smaller set (red) and kept the larger set (black). The duplication time was 1, 2, 3, and 4 (as depicted in the x-axis). (**B**) Strategy 2: Random under-sampling disregarded some data in the larger set (red) and kept the smaller set (black).

**Figure 4 jpm-12-01377-f004:**
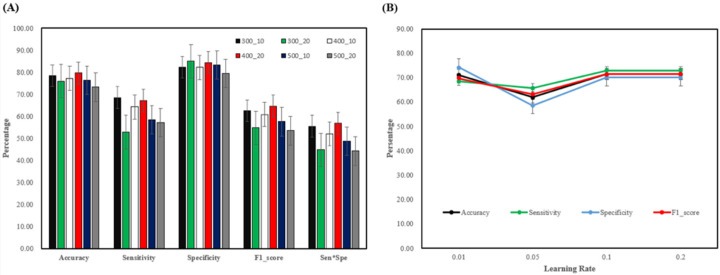
(**A**) The first number (300, 400, 500) represents the n_estimators and the second number represents the max_depth (10, 20). Analysis of this cross pair shows that the most stable performance is when the hyperparameter value is 400_20 (**B**) The learning rate is the lowest at 0.05 and 0.1, 0.2 are more stable.

**Figure 5 jpm-12-01377-f005:**
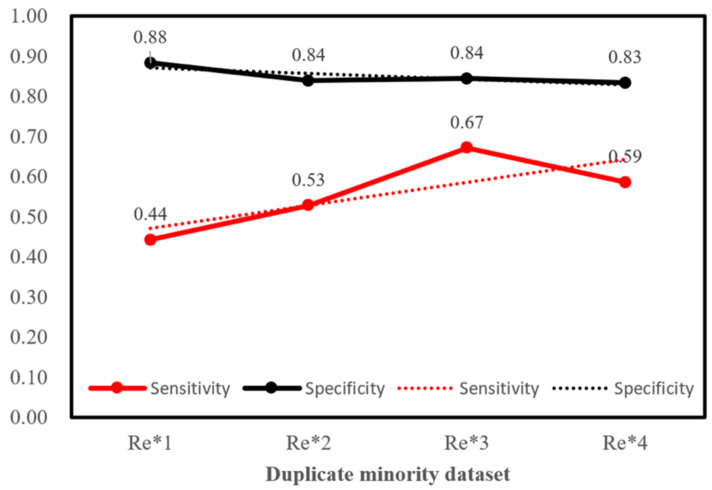
The minority dataset was replicated several times in order to reduce data imbalance, and after the data was replicated three times, the difference in the number of recurring(re) and non-recurring data was minimal. Although it slightly reduced specificity, it averaged out recurrence data and greatly improved sensitivity.

**Figure 6 jpm-12-01377-f006:**
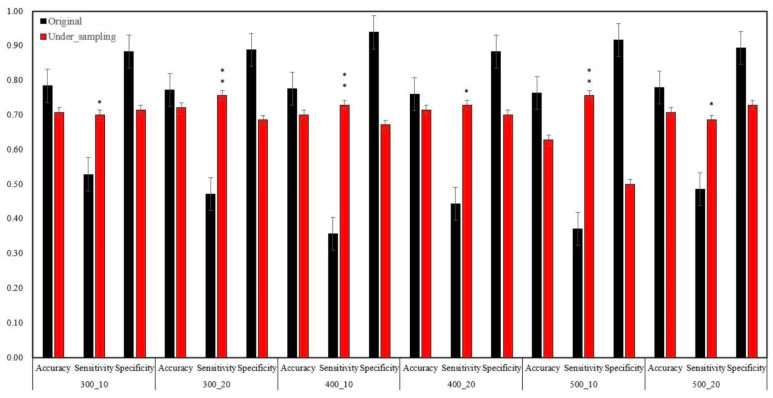
Different hyperparameters with random under-sampling strategy. * significant, ** highly significant.

**Figure 7 jpm-12-01377-f007:**
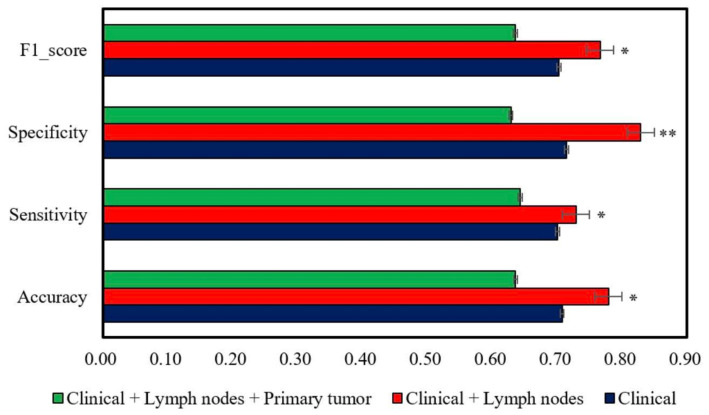
The efficiency of accuracy, sensitivity, and specificity values is the highest when using the combination of clinical data and lymph nodes to predict neck recurrence. * significant, ** highly significant.

**Figure 8 jpm-12-01377-f008:**
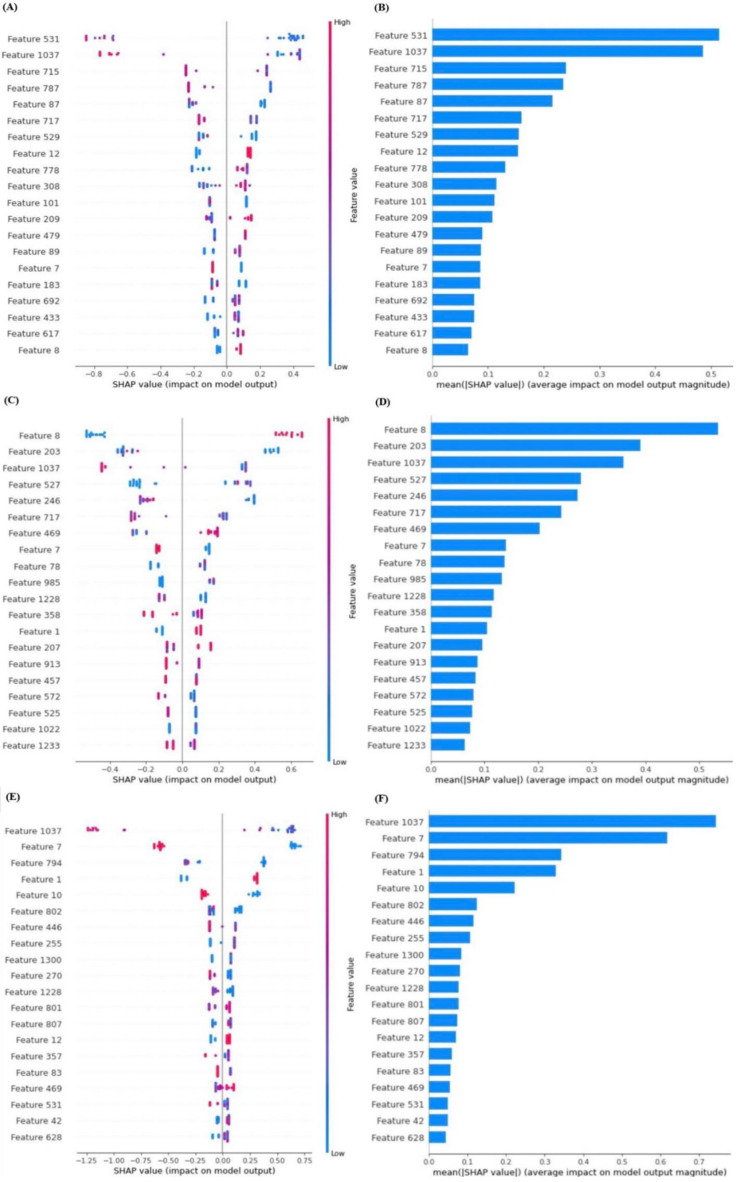
The feature importance of the XGBoost model after predicting neck recurrence. The feature names can be found in Table A1. (**A**) The SHAP value of the first model (**B**) The mean SHAP value of the first model (**C**) The SHAP value of the second model (**D**) The mean SHAP value of the second model (**E**) The SHAP value of the third model (**F**) The mean SHAP value of the third model.

**Table 1 jpm-12-01377-t001:** Patient Characteristics (Total = 79).

Characteristics	Value
Age	37–78 years (Median, 51)
Sex	Male: 78 (99%), Female: 1 (1%)
Smoking	Yes: 68 (86%), no: 11 (14%)
Betel nut squid	Yes: 52 (66%), no: 27 (34%)
Alcoholism	Yes: 52(66%), no: 27 (34%)
Primary lesion site	
Oropharynx	36 (46%)
Hypopharynx	43 (54%)
Recurrence	
Local relapse	Yes: 30 (38%), no: 49 (62%)
Neck relapse	Yes: 22 (28%), no: 57 (72%)
Distant metastasis	Yes: 13 (16%), no: 66 (84%)
SUVmax of primary tumor	Max: 30.6, min:2.2
SUVmax of lymph node	Max: 28.5, min:1.3
T-stage	T1:4 (5%) T2:30 (38%) T3:19 (24%) T4:26 (33%)
N-stage	N0:3 (4%) N1:14 (18%) N2:58 (73%) N3:4(5%)
Max Diameter of LN (cm)	Max:10.8, min:0.8
Existence of necrotic lymph node	Yes: 40 (51%), no: 39 (49%)

SUVmax, maximum standard uptake value.

**Table 2 jpm-12-01377-t002:** Comparison of Random Forest and XGBoost Values. The values of the subsamples (0.5, 0.8, 1). The n_estimators (400)_ max_depth (20)_ learning rate (0.1) are the same in each subsample.

Algorithm	Accuracy	Sensitivity	Specificity	AUC
Random Forest	0.61 ± 0.11	0.59 ± 0.32	0.63 ± 0.20	0.71
XGBoost (0.5) *	0.64 ± 0.06	0.70 ± 0.14	0.57 ± 0.21	0.73
XGBoost (0.8) *	0.71 ± 0.09	0.73 ± 0.13	0.70 ± 0.13	0.74
XGBoost (1.0) *	0.72 ± 0.10	0.76 ± 0.22	0.69 ± 0.11	0.79

* The number indicates the parameter ‘subsample’, which is a ratio of using data to build the main tree in XGBoost. AUC, Area Under the Curve.

**Table 3 jpm-12-01377-t003:** Performance of over-sampling Data.

Hyperparameters	Original Ratio of Data				Duplicate Minority Data Three Times			
N_Estimators_Max_Depth	Accuracy	Sensitivity	Specificity	F1_Score	Accuracy	Sensitivity	Specificity	F1_Score
300_10	0.78 ± 0.05	0.53 ± 0.19	0.88 ± 0.05	0.56 ± 0.15	0.78 ± 0.06	0.69 ± 0.22	0.82 ± 0.08	0.63 ± 0.14
300_20	0.77 ± 0.06	0.47 ± 0.11	0.89 ± 0.08	0.54 ± 0.09	0.76 ± 0.05	0.53 ± 0.13	0.85 ± 0.06	0.55 ± 0.10
400_10	0.78 ± 0.06	0.36 ± 0.20	0.94 ± 0.05	0.45 ± 0.21	0.77 ± 0.05	0.64 ± 0.15	0.82 ± 0.09	0.61 ± 0.08
400_20	0.76 ± 0.07	0.44 ± 0.22	0.88 ± 0.06	0.49 ± 0.19	0.80 ± 0.05	0.67 ± 0.11	0.84 ± 0.05	0.65 ± 0.09
500_10	0.76 ± 0.06	0.37 ± 0.13	0.92 ± 0.08	0.46 ± 0.14	0.76 ± 0.10	0.59 ± 0.21	0.83 ± 0.11	0.58 ± 0.17
500_20	0.78 ± 0.06	0.49 ± 0.19	0.89 ± 0.09	0.54 ± 0.15	0.73 ± 0.05	0.57 ± 0.18	0.79 ± 0.09	0.54 ± 0.10

**Table 4 jpm-12-01377-t004:** Performance of under-sampling data.

The Same Ratio of Data				
N_Estimators_Max_Depth	Accuracy	Sensitivity	Specificity	F1_Score
300_10	0.71 ± 0.07	0.70 ± 0.15	0.71 ± 0.11	0.70 ± 0.09
300_20	0.72 ± 0.08	0.76 ± 0.09	0.69 ± 0.12	0.73 ± 0.08
400_10	0.70 ± 0.11	0.73 ± 0.17	0.67 ± 0.19	0.70 ± 0.12
400_20	0.71 ± 0.09	0.73 ± 0.13	0.70 ± 0.13	0.71 ± 0.10
500_10	0.63 ± 0.13	0.76 ± 0.18	0.50 ± 0.16	0.67 ± 0.13
500_20	0.71 ± 0.09	0.69 ± 0.14	0.73 ± 0.16	0.70 ± 0.09

## Data Availability

Not applicable.

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
