# Peer review of "Neck Lymph Node Recurrence in HNC Patients Might Be Predicted before Radiotherapy Using Radiomics Extracted from CT Images and XGBoost Algorithm"

_jpm, 2022, doi:10.3390/jpm12091377_

Round 1
Reviewer 1 Report
The authors have prevented a commendable study regarding the application of artificial intelligence in head and cancer management. Here are a few notes aimed at improving the manuscript.
Title:
The authors aimed at evaluating the effect of under-sampling and over-sampling on sensitivity. Both under-sampling and over-sampling are known techniques to handle imbalanced datasets. Therefore, these techniques definitely have an effect on important performance metrics such as sensitivity. Hence, it is important to properly justify the novelty and interest of this paper to the readership of this journal.
Abstract:
The performance of the model before under-sampling and over-sampling should be mentioned. Thus, the effects of the under-sampling and over-sampling can easily be evaluated by the reader. In addition, the under-sampling or over-sampling techniques should be mentioned.
Methodology
The composition of the data largely includes 36 patients with oropharyngeal cancer (OPC) and 43 patients with hypopharyngeal cancer (HPC). Therefore, can this still be considered viable for HNC where there was a lack of many HNC subsites in the data composition? How will this model perform if evaluated on new cases of tongue squamous cell carcinoma?
The methodology is logical. However, the model may not be generalized for HNC. Rather, it may be limited to OPC and HPC.
Obviously, there is also a case of gender bias in terms of the data used in this study. Therefore, it would be worthwhile to see how gender affects the target variable in this study.
Discussion
The paragraphs in the discussion need to be properly arranged to discuss the results obtained in this study. In this present form, it is a bit difficult to follow up with the discussion. The discussion in this present form is not discussing the results and previous studies. Thus, it becomes challenging to see the contribution of this study.
Author Response
Point1 - Title:
The authors aimed at evaluating the effect of under-sampling and over-sampling on sensitivity. Both under-sampling and over-sampling are known techniques to handle imbalanced datasets. Therefore, these techniques definitely have an effect on important performance metrics such as sensitivity. Hence, it is important to properly justify the novelty and interest of this paper to the readership of this journal.
Response 1: We have modified the title of this paper as follows: “A Study in Predicting Neck Lymph Nodes Recurrence in Patients with Head and Neck Cancer Based on an Unbalanced Dataset”
Point2 - Abstract:
The performance of the model before under-sampling and over-sampling should be mentioned. Thus, the effects of the under-sampling and over-sampling can easily be evaluated by the reader. In addition, the under-sampling or over-sampling techniques should be mentioned.
Response 2: Thank you for your comments and suggestions. The original dataset performed as follows: accuracy=0.76±0.07, sensitivity=0.44±0.22, specificity=0.88±0.06. After we used the over-sampling technique, the accuracy, sensitivity, specificity values were 0.80±0.05, 0.67±0.11, 0.84±0.05, respectively. Furthermore, after using the under-sampling technique, the accuracy, sensitivity, specificity values were 0.71±0.09, 0.73±0.13, 0.70±0.13, respectively. The suggested changes have already been added to the manuscript.
Point3 - Methodology:
The composition of the data largely includes 36 patients with oropharyngeal cancer (OPC) and 43 patients with hypopharyngeal cancer (HPC). Therefore, can this still be considered viable for HNC where there was a lack of many HNC subsites in the data composition? How will this model perform if evaluated on new cases of tongue squamous cell carcinoma?
The methodology is logical. However, the model may not be generalized for HNC. Rather, it may be limited to OPC and HPC.
Response 3: In the era of artificial intelligence, this study aimed to use PET/CT-derived radiomics for predicting radiotherapy-based outcome in patients with head and neck cancer. Because definitive chemoradiotherapy/radiotherapy is a mainstay of treatment for patients with cancers of hypopharyx, larynx, and oropharynx, our goal is to optimize the individualized treatment decision between organ preservation and treatment modification. Of course, our findings need external validation. The base of the tongue is generally categorized as oropharyngeal cancer, whereas mobile tongue cancer belongs to cancer of oral cavity, and patients are always treated with surgery with or without adjuvant radiotherapy. In patients with buccal or lip cancer, radiotherapy is mainly used for adjuvant treatment.
Point 4: Obviously, there is also a case of gender bias in terms of the data used in this study. Therefore, it would be worthwhile to see how gender affects the target variable in this study.
Response 4: Regarding the gender bias in this study, we would like to explain that the incidence of human papilloma virus (HPV)-associated oropharyngeal cancer was very low in many Asian countries like Taiwan (<20% in our study cohort). More than 90% of our studied patients had a history of smoking, alcoholism, or betel nut squid, which is regarded a carcinogen of their cancers. On the other hand, the incidence of HPV-associated hypopharyngeal cancer was also very low in Asian countries (<10% in our study cohort). According to our cultural habit, very few women had a history of smoking, alcoholism, or betel nut chewing, this was the reason that the gender for most of the studied objects was male
Point 5: Discussion
The paragraphs in the discussion need to be properly arranged to discuss the results obtained in this study. In this present form, it is a bit difficult to follow up with the discussion. The discussion in this present form is not discussing the results and previous studies. Thus, it becomes challenging to see the contribution of this study.
Response 5: Thank you for your comment. The paragraphs have been re-arranged to make the chapter more comprehensive. We also discuss the results and previous studies.
Reviewer 2 Report
Tsai et al. report CT radiomics in head neck malignancies using over-sampling and under-sampling techniques. Kindly find my comments below
Comments:
1. Include details of CT acquisition parameters and machine details in the methodology.
2. Will suggest using “primary tumor/ disease” instead of “tumor in situ”
3. To elaborate on image preprocessing, details of radiomics feature
4. To mention who had done the image segmentation eg. Radiologist? (xx years of experience)
5. To mention in selection criteria if all patients were required to have minimal follow up duration
6. To specify validation techniques
7. Authors need to present classifier performances using lymph node data alone since clinical data alone has good results and values went down when combined features were used from primary, nodes along with clinical data
Author Response
Point 1: Include details of CT acquisition parameters and machine details in the methodology.
Response 1: The patients were scanned using a PET/CT scanner (PET/CT-16 slice, Discovery STE; GE Medical System, Milwaukee, WI, USA). The patients were instructed to fast for at least 4 hours before the administration of 18F-FDG, and FDG PET/CT imaging was conducted approximately one hour after the administration of 370 MBq of 18F-FDG. The details of CT image acquisition has been described in detail in chapter 2.2.
Point 2: Will suggest using “primary tumor/ disease” instead of “tumor in situ”
Response 2: Thank you for your suggestion. We have replaced “tumor in situ” with “primary tumor” in our manuscript.
Point 3: To elaborate on image preprocessing, details of radiomics feature
Response 3:
The part describing radiomics has been included in chapter 2.4 and is as follows:
The term radiomics was first coined by Lambin et al. 2012, to describe quantitative medical imaging data. Radiomics involves extracting high-throughput data from medical images such as CT, PET, MRI or SPECT scans through advanced mathematical and statistical analysis. Radiomic has several types of features. These are Shape-based features, First-order statistics, Gray Level Run Length Matrix (GLRLM), Gray Level Co-Occurrence Matrix (GLCM), Gray Level Size Zone Matrix (GLSZM), Neighboring Gray Tone Difference Matrix (NGTDM), Gray Level Dependence Matrix (GLCM), A Laplacian of Gaussian (LoG) features and Wavelet features. Radiomics explores the intensity, shape and texture of the tumor in order to calculate thousands of advanced features.
Point 4: To mention who had done the image segmentation eg. Radiologist? (xx years of experience)
Response 4: The segmentation of the images was done by members of our team and the processed images were then approved by a physician with more than 30 years of experience.
Point 5: To mention in selection criteria if all patients were required to have minimal follow up duration
Response 5: The minimal follow up duration for patients was 12 months.
Point 6: To specify validation techniques
Response 6: After utilizing the over-sampling technique and under-sampling techniques, we used the XGBoost algorithm to analyze the data which was separated into a training set (70%) and test set (30%). The analysis was run one hundred times, each time with randomly chosen sets of patients. Due to the small cohort size, we didn’t have a validation set, therefore, the test data was used to the validate the algorithm.
Point 7: Authors need to present classifier performances using lymph node data alone since clinical data alone has good results and values went down when combined features were used from primary, nodes along with clinical data
Response 7: Local recurrence is defined as a cancer that recures in the same place as the original cancer or very close to it. Regional recurrence (neck recurrence in head and neck) means that the tumor has grown into lymph nodes or tissues near the original cancer (National Cancer institution https://www.cancer.gov/types/recurrent-cancer). Therefore, we used lymph nodes to predict neck recurrence. When we added the primary tumor data (used for predicting local recurrence) the performance in predicting neck recurrence decreased.
Reviewer 3 Report
general comments:
* low cohort size: the 2022 edition of the HECKTOR data challenge has 489 patients with recurrence free survival data and PET-CT images and will most likely be more impactful. Investigation should be made to see if this dataset can be used for your research, to validate your pipeline.
* lengthy details of machine learning algorithm tuning: these details are moderately relevant for a medical journal and its target audience.
* not enough clarification on the choice of machine learning algorithm: why use xgboost when other (simpler) algorithms could be used ? You stated that random forest might be worse, but it would be easy to check it directly on your dataset and compare different algorithms performance using cross validation on the training set.
* the choice of the metrics do not correspond to the censored nature of your data. Recurrence is not a simple classification task, it requires to take into account censorship. As such, classification metrics are not relevant, and C index should be used to evaluate model performance.: your problem should be reframe as a survival task rather than just a classification task.
* it is unclear if the performance reported are for the training set or the test set. Moreover, the xgboost algorithm uses a validation set to monitor the overfitting during training, like deep learning algorithms. It should be clarified if the test set has been used or not as a validation set for xgboost. If the case, all results are most likely over optimist.
materials and methods:
- this paragraph should not detail the patient characteristics but how the cohort has been collected. It is unclear why do we go from 110 to 79 patients in the selection process.
- why not use the PET images to extract radiomics, which are most likely more relevant for RN/LR/DM ?
- table 1:
* please provide percentage rather than raw values
* max diameter of LN: in which unit ? cm I think.
* define Necrotic LN
Results
section 3.1 : "We found that radiomics derived from CT was superior 159 than that of PET images for predicting the neck node recurrence". This analysis was not described in the M&M section where you stated you will only use CT radiomics.
Figures:
* figure 8 : feature importance when feature name is not very self describing does not add value to this research.
Author Response
Point 1: low cohort size: the 2022 edition of the HECKTOR data challenge has 489 patients with recurrence free survival data and PET-CT images and will most likely be more impactful. Investigation should be made to see if this dataset can be used for your research, to validate your pipeline.
Response 1: Thank you for your suggestion. We have requested access to the open data set images from Dr.Vallières (the Canadian data set resource of HECKTOR data challenge) on March 2022 at https://www.cancerimagingarchive.net/. After we received the information, we found out the data set has three outcomes: Local recurrence, Distant Metastasis and Death. However, in our research, our end point is neck (lymph nodes) recurrence which wasn’t recorded in the HECKTOR data challenge clinical data. Hence, due to the different end point, we couldn’t use the dataset to validate our results.
Point 2: lengthy details of machine learning algorithm tuning: these details are moderately relevant for a medical journal and its target audience.
Response 2: Thank you for your suggestion. We edited the text to properly reflect the tuning process in chapter 3.2
Point 3: not enough clarification on the choice of machine learning algorithm: why use xgboost when other (simpler) algorithms could be used ? You stated that random forest might be worse, but it would be easy to check it directly on your dataset and compare different algorithms performance using cross validation on the training set.
Response 3: Thank you for your advice. We originally did use the Random Forest algorithm; however, the value of accuracy was 0.61, sensitivity=0.59, specificity = 0.63, which were unsatisfactory. Therefore, we opted to use the XGBoost algorithm instead which offered better results. The values when using Random Forest can be seen in Table 2 which we included to showcase the difference.
Point 4: the choice of the metrics do not correspond to the censored nature of your data. Recurrence is not a simple classification task, it requires to take into account censorship. As such, classification metrics are not relevant, and C index should be used to evaluate model performance.: your problem should be reframe as a survival task rather than just a classification task.
Response 4: We discussed this topic with an experienced clinical physician, and the reason why we decided to conduct research into recurrence rather than survival was because early recognition of patients at risk for nodal failure after curative nonsurgical treatment can optimize the individual treatment schemes by reducing the number of patients undergoing unsuitable treatment. Therefore, we focused on whether there will be recurrence rather than survival. This way, patients can be treated alternatively.
Point 5: it is unclear if the performance reported are for the training set or the test set. Moreover, the xgboost algorithm uses a validation set to monitor the overfitting during training, like deep learning algorithms. It should be clarified if the test set has been used or not as a validation set for xgboost. If the case, all results are most likely over optimist.
Response 5: Due to the small cohort of our patients, we only have training and test data sets. The test data was used as a validation set for the algorithm.
Point 6: materials and methods: this paragraph should not detail the patient characteristics but how the cohort has been collected. It is unclear why do we go from 110 to 79 patients in the selection process
Response 6: There are two reasons why we downsized the patient population from 110 to 79. First, the CT/PET images of 17 patients couldn’t be loaded in the Lifex segmentation software. This was most likely due to an incompatible format. We attempted to obtain the data with the compatible format, however, the data we received was the only data available. Second, 14 patients only had primary tumors, with no obvious lymph nodes that could be seen in the images.
Point 7: why not use the PET images to extract radiomics, which are most likely more relevant for RN/LR/DM ?
Response 7: The reason for choosing CT images is touched upon in chapter 3.1. After using the 8 extractors from radiomics to extract features from the 18F-FDG PET/CT images which were used to predict the neck recurrence, we found that, by an-alyzing PET images with lymph node masks there were 105 parameters in total with p-value < 0.005. By analyzing the CT images with lymph node masks, we found 258 parameters in total with p-value < 0.005. We found that radiomics derived from CT was superior than that of PET images for predicting the neck node recurrence. There-fore, the subsequent analysis used the CT scans with image features extracted by the exampleCT extractor combined with the clinical data for machine learning.
Point 8: table 1:
* please provide percentage rather than raw values
* max diameter of LN: in which unit ? cm I think.
* define Necrotic LN
Response 8: Thank you for your suggestions. All three points have been revised.
Point 9: Results - section 3.1 : "We found that radiomics derived from CT was superior 159 than that of PET images for predicting the neck node recurrence". This analysis was not described in the M&M section where you stated you will only use CT radiomics.
Response 9: Thank you for your comment. The selection process for the images has been described and included in chapter 2.2.
Point 10: Figures - figure 8 : feature importance when feature name is not very self describing does not add value to this research.
Response 10: A list of significant features with their names was added. It can be found in the Appendix.
|
|
Top 10 Features of Figure 8. (A)(B) |
|
Feature 531 |
original_firstorder_Median |
|
Feature 1037 |
wavelet-LHH_glszm_GrayLevelNonUniformityNormalized |
|
Feature 715 |
wavelet-HHL_firstorder_Mean |
|
Feature 787 |
wavelet-HHL_glszm_SizeZoneNonUniformityNormalized |
|
Feature 87 |
diagnostics_Mask-interpolated_Mean |
|
Feature 717 |
wavelet-HHL_firstorder_Median |
|
Feature 529 |
original_firstorder_Mean |
|
Feature 12 |
N-SUVmax >4.9 |
|
Feature 778 |
wavelet-HHL_glszm_GrayLevelNonUniformity |
|
Feature 308 |
log-sigma-3-0-mm-3D_gldm_DependenceVariance |
|
|
Top 10 Features of Figure 8. (C)(D) |
|
Feature 8 |
lesion site 1)Opx 2)HPx |
|
Feature 203 |
log-sigma-2-0-mm-3D_glcm_DifferenceAverage |
|
Feature 1037 |
wavelet-LHH_glszm_GrayLevelNonUniformityNormalized |
|
Feature 527 |
original_firstorder_Kurtosis |
|
Feature 246 |
log-sigma-2-0-mm-3D_glrlm_ShortRunEmphasis |
|
Feature 717 |
wavelet-HHL_firstorder_Median |
|
Feature 469 |
log-sigma-5-0-mm-3D_glcm_InverseVariance |
|
Feature 7 |
alcohol 1) Yes 2) No |
|
Feature 78 |
diagnostics_Image-interpolated_Size |
|
Feature 985 |
wavelet-LHH_glcm_ClusterProminence |
|
|
Top 10 Features of Figure 8. (E)(F) |
|
Feature 1037 |
wavelet-LHH_glszm_GrayLevelNonUniformityNormalized |
|
Feature 7 |
alcohol 1) Yes 2) No |
|
Feature 794 |
wavelet-HLH_firstorder_10Percentile |
|
Feature 1 |
diagnosis: 1) OPC; 2) HPC |
|
Feature 10 |
SUVmax |
|
Feature 802 |
wavelet-HLH_firstorder_MeanAbsoluteDeviation |
|
Feature 446 |
log-sigma-5-0-mm-3D_firstorder_Median |
|
Feature 255 |
log-sigma-2-0-mm-3D_glszm_LargeAreaLowGrayLevelEmphasis |
|
Feature 1300 |
wavelet-LLL_glszm_LargeAreaLowGrayLevelEmphasis |
|
Feature 270 |
log-sigma-3-0-mm-3D_firstorder_Kurtosis |
Round 2
Reviewer 1 Report
The authors have addressed most of the comments. Although, the rationale behind some of the responses may lead to further concerns by the readership of this journal.
Minor comments:
1. The title of this paper should relate to the stated objectives of the study as mentioned in the introduction. In the present form, there seems no correlation between the title and the objectives.
2. I strongly think that the composition of the dataset used in this study does not qualify to be addressed as head and neck cancer. Therefore, the limitations in terms of the absence of other subsites should be mentioned in the study.
3. Handling imbalance dataset with undersampling or oversampling is not novel. Handling imbalanced data should be standard practice when dealing with an imbalanced dataset. A study that would revolve around a new technique for handling data imbalance would be worthwhile. However, in the present form of this study, it is still uncertain what this study would add to the readership of this journal.
4. Some of these points may be mentioned as part of the limitations so that the readers can clearly understand this study.
Author Response
Minor comments:
Point 1: The title of this paper should relate to the stated objectives of the study as mentioned in the introduction. In the present form, there seems no correlation between the title and the objectives.
Response 1: Thank you for your suggestion. We have modified the title of this paper as follows:” Neck Lymph Node Recurrence of HNC Patients Might Be Predicted Before Radiotherapy - Using Radiomics Extracted From CT Images and XGBoost Algorithm”.
Point 2: I strongly think that the composition of the dataset used in this study does not qualify to be addressed as head and neck cancer. Therefore, the limitations in terms of the absence of other subsites should be mentioned in the study.
Response 2: Thank you very much for your suggestion. We have added the following section where we discuss this limitation to the discussion.
“This study had some limitations. First, external validation with independent data sets is essential to create the model’s clinical utility because this study was conducted at a single institute. Second, although organ preservation with definitive radiotherapy is a mainstay of treatment for patients with cancers of hypopharynx or oropharynx, our findings should be interpreted cautiously because of the absence of other subsites of head and neck cancers. Nonetheless, this study represents a step toward enabling customization of precision therapy for head and neck cancer patients when the optimal management of the metastatic neck node becomes an issue of debate. After further validation and inclusion of more tumor sites, oncologists may use the proposed model to advise patients on the relative suitability of treatment options.”
Point 3: Handling imbalance dataset with undersampling or oversampling is not novel. Handling imbalanced data should be standard practice when dealing with an imbalanced dataset. A study that would revolve around a new technique for handling data imbalance would be worthwhile. However, in the present form of this study, it is still uncertain what this study would add to the readership of this journal.
Response 3: Thank you for your comment. The under-sampling and over-sampling techniques were part of this study, however, the main propose of this study was to see how information gathered from lymph nodes, rather the primary tumors, could be used to predict regional recurrence (neck recurrence in HNC). We found that the imbalanced data strategy improved the results, however, the strategy itself was not the focus of this study. The main focus of this study was to show the importance of indicators of personal treatment. The outcome of metastatic neck nodes in patients with HNC receiving radiotherapy for organ preservation can be predicted based on the results of machine learning. This way, patients can receive personalized treatment.
We modified several parts of the paper to highlight the main objective of this study.
Point 4: Some of these points may be mentioned as part of the limitations so that the readers can clearly understand this study.
Response 4: Thank you for your suggestion. We addressed your points by modifying several parts of the paper, including the title, discussion, and conclusion.
Reviewer 2 Report
Thank you for answering my comments
Author Response
Dear Reviewer,
Thank you very much for your insightful suggestions. I am glad I was able to succesfully address your comments.
Kind regards
Yi-Lun Tsai
Reviewer 3 Report
Response 1: thanks for the clarification.
Response 2: it is indeed much better.
Response 3: The clarification is nice, but the reason xgboost is better is overfit. During the training of xgboost you have to watch the validation set, and stop at an appopriate timing when the performance is best. For this reason, you may have overfit to your validation set. Lack of test set is critical for such algorithms, whereas not so much for random forest where you can use out of bag estimation of performance on the train set without "watching" your validation set for performance. There is indeed usually a better performance of boosted trees vs random forest, but not that much.
Response 4: You missed the point here. It was not about local recurrence vs PFS, it was about the censored nature of your data. You cannot use simple classification when there is censored data, you have to use algorithm that take into account the censorship. I strongly suggest you drop completely xgboost, and use random survival forest (you can find a nice package in R that will do just that: ranger).
Response 5: This is a critical design flaw when using xgboost. cf Response 3 and 4 for improvement.
Response 6: This is a bit disappointing. There are lot of software out there that can be used to extract radiomics. I suggest you use Pyradiomics python package for this task. The removal of 17 patients due to software not working is not likely to induce any bias in the results, but what will happen in real life if you cannot extract the radiomics for a specific patient ?
Response 7: Ok
Response 8: define Necrotic LN not adressed.
Response 9: adressed
Response 10: adressed
Additionnal point: regarding clarification of PET CT acquisition, it is very strange that all patients were injected with the same FDG activity (370MBq). The dosage is weight dependant (usually between 2.5 and 3.5 MBq/kg).
